**RESEARCH**

# Host adaptation in gut Firmicutes is associated with sporulation loss and altered transmission cycle

Hilary P. Browne[1*], Alexandre Almeida[2,3], Nitin Kumar[1], Kevin Vervier[1], Anne T. Adoum[2], Elisa Viciani[1], Nicholas J. R. Dawson[1], Samuel C. Forster[1,4,5], Claire Cormie[2], David Goulding[2] and Trevor D. Lawley[1*]

\* Correspondence: hb4@sanger.ac.
uk; tl2@sanger.ac.uk
[1]Host-Microbiota Interactions
Laboratory, Wellcome Sanger
Institute, Hinxton, UK
Full list of author information is
available at the end of the article

## Abstract

**Background:** Human-to-human transmission of symbiotic, anaerobic bacteria is a fundamental evolutionary adaptation essential for membership of the human gut microbiota. However, despite its importance, the genomic and biological adaptations underpinning symbiont transmission remain poorly understood. The Firmicutes are a dominant phylum within the intestinal microbiota that are capable of producing resistant endospores that maintain viability within the environment and germinate within the intestine to facilitate transmission. However, the impact of host transmission on the evolutionary and adaptive processes within the intestinal microbiota remains unknown.

**Results:** We analyze 1358 genomes of Firmicutes bacteria derived from host and environment-associated habitats. Characterization of genomes as spore-forming based on the presence of sporulation-predictive genes reveals multiple losses of sporulation in many distinct lineages. Loss of sporulation in gut Firmicutes is associated with features of host-adaptation such as genome reduction and specialized metabolic capabilities. Consistent with these data, analysis of 9966 gut metagenomes from adults around the world demonstrates that bacteria now incapable of sporulation are more abundant within individuals but less prevalent in the human population compared to spore-forming bacteria.

**Conclusions:** Our results suggest host adaptation in gut Firmicutes is an evolutionary trade-off between transmission range and colonization abundance. We reveal host transmission as an underappreciated process that shapes the evolution, assembly, and functions of gut Firmicutes.

**Keywords:** Sporulation, Intestinal microbiota, Microbiome, Metagenomics, Host adaptation, Genome reduction, Genome evolution, Bacterial transmission, Metabolic specialization

## Background

The human gut is colonized by highly adapted bacteria, primarily from the Firmicutes, Bacteroidetes, Actinobacteria, and Proteobacteria phyla, that are linked with human health and development [1–4]. Co-evolution between humans and these symbiotic, anaerobic bacteria requires that individual bacterial taxa faithfully and efficiently transmit and colonize, as an inability to do either leads to extinction from the indigenous microbiota [5–8]. Key adaptations of symbiotic bacteria in human populations, therefore require coordination of colonization and transmission functions. Gut bacteria must be able to colonize above a certain abundance to achieve sufficient shedding levels to ensure onward transmission, and survive in the environment long enough to encounter a susceptible host [7]. Once ingested, gut bacteria must transit through the gastrointestinal tract, contend with the human immune system, and compete with indigenous bacteria for nutrients and replicative niches to colonize [7, 9].

Transmission of the intestinal microbiota is an ongoing process starting with maternal transmission around birth and continuing throughout life, especially between co-habiting individuals in regular contact [10–20]. In fact, gut symbiont transmission during co-habitation has a stronger effect on the composition of an individual's gut microbiota than human genetics [14], highlighting the importance of transmission in shaping an individual's microbiota composition and functions. Thus, the transmission cycle of gut bacteria is underpinned by a deep evolutionary selection that remains poorly understood.

Spores are metabolically dormant and highly resistant structures produced by Firmicutes bacteria that enhance survival in adverse conditions [21–23]. Sporulation is utilized by anaerobic enteric pathogens such as *Clostridioides difficile* (formerly *Clostridium difficile*) to promote transmission by maintaining environmental viability. Upon ingestion by a new host, the spores germinate in response to intestinal bile acids [24]. We recently demonstrated that at least 50% of the commensal intestinal microbiota also produce resistant spores that can tolerate ambient environmental conditions for weeks and subsequently germinate in response to bile acids [25]. Hence, the production of spores enhances environmental survival promoting host-to-host transmission and colonization for a large proportion of the intestinal microbiota [10, 25–29].

Sporulation is a complex developmental process, dependent on hundreds of genes and takes hours to complete, eventually resulting in the destruction of the original mother cell [23, 30, 31]. As sporulation is integral to the transmission of many gut Firmicutes, we hypothesized that phenotypic loss would confer an advantage linked to an altered transmission cycle no longer reliant on environmental persistence. Loss of sporulation has been demonstrated in experimental conditions under relaxed nutrient selection pressures, indicating maintenance of the phenotype as long as it is beneficial [32, 33]. In this study, combining large-scale genomic analysis with phenotypic validation of human gut bacteria from the Firmicutes phylum, we show that sporulation loss is associated with signatures of host adaptation such as genome reduction and more specialized metabolic capabilities. Human population-level metagenomic analysis reveals bacteria no longer capable of sporulation are more abundant in individuals but less prevalent compared to spore-formers, suggesting increased colonization capacity and reduced transmission range are linked to host adaptation within the human intestinal microbiota.

## Results

### Prediction of sporulation capability in gut Firmicutes

We collated 1358 non-redundant, whole-genome sequences of Firmicutes bacteria derived from different human body sites and other environments (Additional file 1: Fig. S1, Additional file 2: Table S1). In addition, we included 72 genomes from non-sporulating bacteria from Actinobacteria, Bacteroidetes, and Proteobacteria phyla for comparative purposes. Using this collection, we next assigned the presence of 66 sporulation-predictive genes identified using a previously developed machine learning model based on analysis of nearly 700,000 genes and 234 genomes from bacteria with an ethanol sensitive or ethanol resistant phenotype, cultured from human fecal samples [25] (Additional file 2: Table S1). Genes in this sporulation signature include characterized sporulation-associated genes, characterized genes not previously associated with sporulation, and uncharacterized genes that have subsequently been demonstrated to be sporulation-associated [34].

Examining the genomes in different bacterial families, we observe three different trends in sporulation signature score (presence of 66 sporulation-predictive genes as a percentage). Families either contain genomes that cluster together with a high sporulation signature score, a low sporulation signature score or a bimodal pattern with both high and low scoring sporulation signature score genomes clustering separately (Fig. 1a, Additional file 2: Table S1). We also observe a strong association between the presence and absence of *spo0A*, the master regulator gene essential for sporulation and genomes clustering with either a high or low sporulation signature score. In total, 98.9% (1343 of 1358) of the genomes are either high scoring for sporulation signature score and contain *spo0A* or are low scoring and lack *spo0A* (Additional file 2: Table S1).

As bacterial sporulation is believed to have evolved once, early in Firmicutes evolution [21, 35, 36], we classify genomes within low scoring sporulation signature clusters as Former-Spore-Formers (FSF) (i.e., Firmicutes that have lost the capability to produce spores) and genomes within high scoring sporulation signature clusters as Spore-Formers (SF). This classification is a refinement of our previously established sporulation signature as it accounts for the different sporulation machinery between taxonomically different bacterial families (see the "Methods" section) [25]. Based on our classification system, the largest families with a bimodal pattern are the host-associated *Erysipelotrichaceae*, *Peptostreptococcaceae*, *Clostridiaceae*, *Ruminococcaceae*, and *Lachnospiraceae* families that are known to contain spore-forming bacteria (Fig. 1a). Importantly, in bacterial families that are known not to produce spores, like *Lactobacillaceae* and *Streptococcaceae* within the Firmicutes, we only observe genomes we classify as FSF. Furthermore, other bacteria from the Bacteroidetes, Actinobacteria, and Proteobacteria phyla which do not make spores are also classified as FSF (P<0.0001, Mann-Whitney, sporulation signature score comparison between genomes of SF, FSF, and non-Firmicutes) (Fig. 1a, Additional file 1: Fig. S2, Additional file 2: Table S1).

### Loss of sporulation genes in gut Firmicutes

To investigate the genetic processes and selective forces underlying sporulation loss in human gut symbionts, we combined genomes from gut-associated SF and FSF bacteria

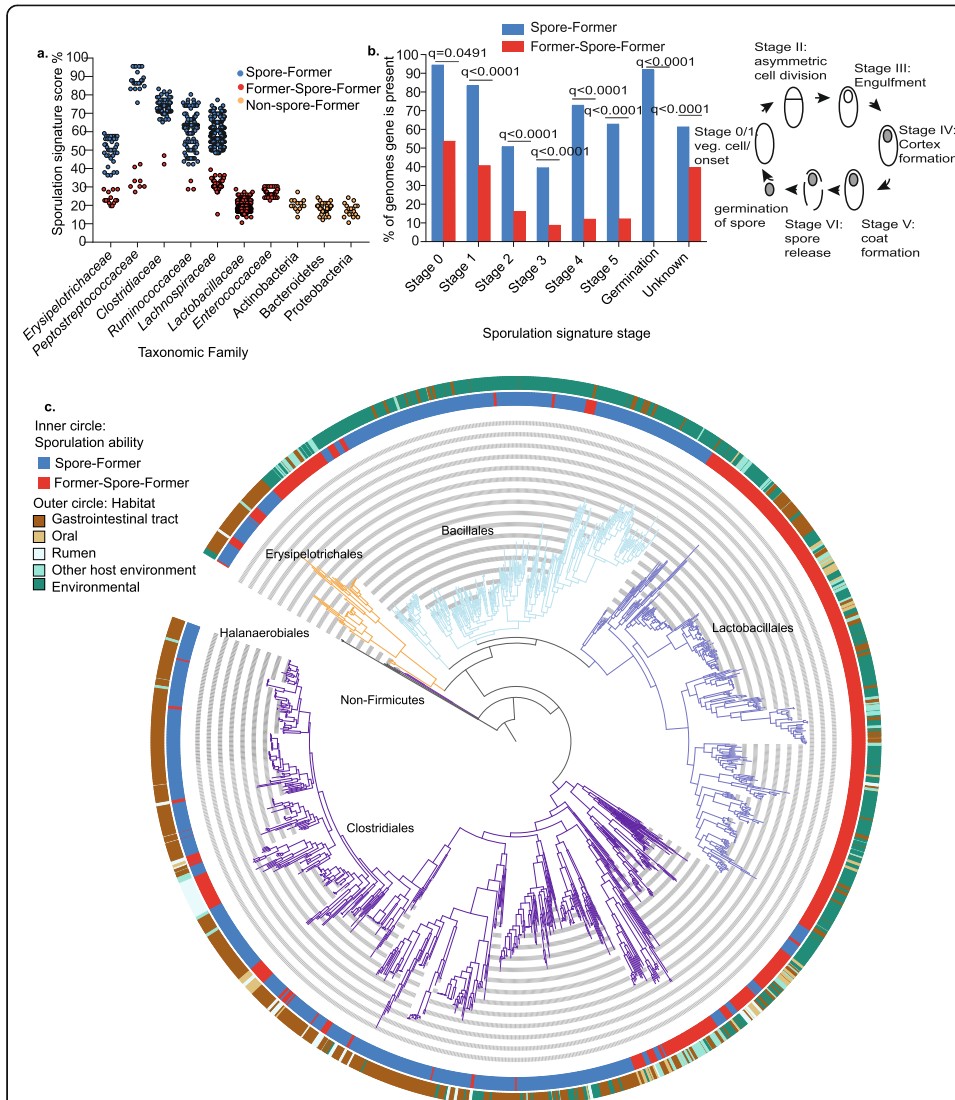

**Fig. 1** Sporulation loss in distinct evolutionary lineages of gut Firmicutes. **a** Prediction of sporulation capability in human-associated Firmicutes families based on the presence of 66 sporulation-associated genes. *Erysipelotrichaceae*, *Peptostreptococcaceae*, *Clostridiaceae*, *Ruminococcaceae*, and *Lachnospiraceae* families have a bimodal pattern with genomes either having a high scoring (blue dots) (classified as Spore-Formers) or a low scoring sporulation signature score (red dots) (classified as Former-Spore-Formers). *Lactobacillaceae*, *Enterococcaceae*, Actinobacteria, *Bacteroidetes*, and Proteobacteria are all non-spore-forming (yellow dots) and contain low scoring genomes that lack *spo0A* which is essential for sporulation. **b** Presence and absence of sporulation signature genes were determined in gut-associated Spore-Formers (SF) (n=456) and Former-Spore-Formers (FSF) (n=117) bacteria. FSF have less sporulation signature genes for all sporulation stages compared to SF (all stages q< 0.0001, except for stage 0 q=0.0491, Fisher's exact test, adjusted for multiple testing). The cartoon describes sporulation stages. **c** Phylogeny of the Firmicutes constructed from 40 universal protein-coding genes extracted from 1358 whole-genome sequences. Sporulation has been lost at large taxonomic scales (*Lactobacillales* order) and at small taxonomic scales (within host-associated *Erysipelotrichales* and *Clostridiales* orders). Major taxonomic orders are indicated by branch colors and name, black branches at the base of phylogeny represent non-Firmicutes root derived from Actinobacteria genomes

(SF n=456, FSF n=117), determined the presence of the 66 sporulation signature genes, and assigned them to their respective sporulation stage. As expected, FSF genomes contain fewer sporulation signature genes for each sporulation stage compared to SF

genomes (all stages, q< 0.0001 except for stage 0 (q= 0.0491, Fisher's exact test) (Fig. 1b). Early-stage (stages 0 and I) sporulation genes which are unknown in function or have pleiotropic, non-sporulation-related functions are maintained to a greater degree compared to later-stage sporulation genes in FSF genomes (stage 0 sporulation genes are, on average, present in 53.7% of FSF genomes, stage I present in 40.7% of FSF genomes). Later-stage sporulation signature genes that are sporulation-specific are absent to a greater degree in FSF genomes (stage II sporulation genes are, on average, present in 16.2% of FSF genomes, stage III genes present in 8.8% of FSF genomes, stage IV genes are present in 12% of FSF genomes, stage V genes are present in 12.2% of FSF genomes, and germination stage genes are absent from FSF genomes). Hence, sporulation-specific genes may be lost as there is no advantage in maintaining them.

We next phenotypically validated the lack of sporulation in gut-associated FSF. We exposed cultures of 41 phylogenetically diverse species from 6 different Firmicutes families, SF (n=26) and FSF (n= 15) to 70% ethanol for 4 h and then cultured on YCFA nutrient media with sodium taurocholate to stimulate germination of ethanol-resistant spores [25, 37] (Additional file 2: Table S1). In addition, to account for bacteria that require intestinal signals to produce spores not present in our experimental conditions, we also recorded whether these species were originally cultured from ethanol-exposed fecal samples [25]. Only SF species (12 out of 26) were successfully cultured after ethanol exposure. A further 9 were not cultured after ethanol exposure but were originally isolated from ethanol exposed feces, highlighting that for some species sporulation is not induced in vitro. Taken together, 21 of 26 total (81%) produce ethanol-resistant spores (Additional file 1: Fig. S3a). No FSF survived ethanol exposure (0/15 (0%)), and none were originally isolated from ethanol-exposed feces (Additional file 2: Table S1). Transmission electron microscopy (TEM) imaging of 21 of the 41 species confirmed the presence of spores in spore-forming bacteria only. TEM images of spores from six species representing four different bacterial families are shown in (Additional file 1: Fig. S3b). Thus, we demonstrate loss of sporulation-specific genes leading to an absence of spores in distinct evolutionary lineages of bacteria, creating Former-Spore-Formers.

### Independent loss of sporulation in distinct lineages of Firmicutes

The differences in sporulation gene content within and between families indicate a divergence in sporulation capacity between distinct lineages, raising interesting questions regarding the phylogenetic and evolutionary relationship between sporulating and non-sporulating bacteria. We next generated a core gene phylogeny, of the 1358 Firmicutes genomes, and mapped sporulation capability to better understand the evolution of sporulation in human gut symbionts (Fig. 1c). Our analysis places the non-gut-associated, SF order Halanaerobiales at the base of the phylogeny [38]. Subsequent, large-scale loss of sporulation within taxonomic orders such as the *Lactobacillales* is evident (all 344 genomes are predicted to be FSF), which has been observed before and attributed to adaptation to nutrient-rich habitats [21, 36]. Interestingly, we also observe smaller scale sporulation loss within multiple distinct clades of the host-associated *Erysipelotrichaceae* (26% are FSF), *Peptostreptococcaceae* (26% are FSF), and *Lachnospiraceae* (18% are FSF) families [39]. Within host-associated bacteria, sporulation has been lost to a greater degree in non-gut habitats (96.6% of oral-associated bacteria are

FSF, 84.1% of rumen-associated bacteria are FSF, while only 20.7% of gut-associated bacteria are FSF). Thus, although sporulation is a core function of the human gut microbiota, as it enhances fecal-oral transmission, we reveal loss of sporulation capability in multiple distinct lineages of human-associated Firmicutes bacteria.

### Genome reduction in gut Former-Spore-Formers

Genome reduction is a feature of host adaptation that has been observed in different environments, including the human gut, and is characterized by a loss of genes not required to survive in an ecosystem [40–45]. To determine if loss of sporulation genes in FSF is associated with broader genome decay, we next compared genome sizes of gut FSF and SF bacteria. FSF (n=117) have, on average, genomes that are 36% smaller than SF bacteria (n=456) (P< 0.0001, Mann-Whitney) (Fig. 2a). The same trend is present in FSF genomes in *Erysipelotrichaceae* (38.9% smaller, P<0.0001, Mann-Whitney), *Peptostreptococcaceae* (40.1% smaller, P=0.0002, Mann-Whitney), and *Lachnospiraceae* families (15.6% smaller, P<0.0001, Mann-Whitney) which contain both SF and FSF bacteria (Additional file 1: Fig. S4a). A low genetic redundancy is another feature of host adaptation and is associated with the occupation of stable or constant niches within an ecosystem [43]. Within the same three families, FSF have a lower percentage of paralogous genes in their genomes in comparison to SF bacteria (*Erysipelotrichaceae* P<0.0001, *Peptostreptococcaceae* P=0.0002, and *Lachnopsiraceae* P< 0.0001, Mann-Whitney) (Additional file 1: Fig. S4b). Thus, within FSF bacteria, loss of sporulation genes is associated with broader genome decay, linking an altered transmission cycle with host adaption.

### Metabolic specialization during host-adaptation by gut Former-Spore-Formers

We next carried out functional enrichment analysis to define genome-wide adaptive features differentiating human gut-associated SF and FSF bacteria. In total, 489 genes were enriched in SF and 272 were enriched in FSF. We assigned these genes to functional classes based on their annotation and compared functional classes enriched in both groups (Fig. 2b, Additional file 2: Table S2). Cell motility (P<0.001), amino acid metabolism (P=0.0148), cofactor metabolism (P=0.043), and sporulation (P<0.001, Fisher's exact test) functional classes were statistically significantly enriched in SF compared to FSF. Thus, loss of these functions may be linked to loss of sporulation. No functional class was enriched in FSF compared to SF.

Within SF, we observe a tendency toward biosynthesis of metabolites compared to transport of metabolites in FSF. The majority (31 of 46) of enriched genes with amino acid metabolism functions in SF are biosynthesis-associated (including histidine, methionine, leucine, and isoleucine). By comparison, 6 of 12 enriched genes with amino acid metabolism functions in FSF are transport-associated (only 3 are biosynthesis-associated). Similarly, 41 of 44 enriched genes with cofactor metabolism functions in SF are biosynthesis-associated, including cobalamin (vitamin B12) (n=19 genes of 23 total required for cobalamin biosynthesis) (Additional file 1: Fig. S4c). Cobalamin is primarily acquired by external transportation by gut bacteria and is required for important microbial metabolic processes, including methionine biosynthesis [46, 47]. Species auxotrophic for cobalamin rely on sharing from cobalamin producers, hence these

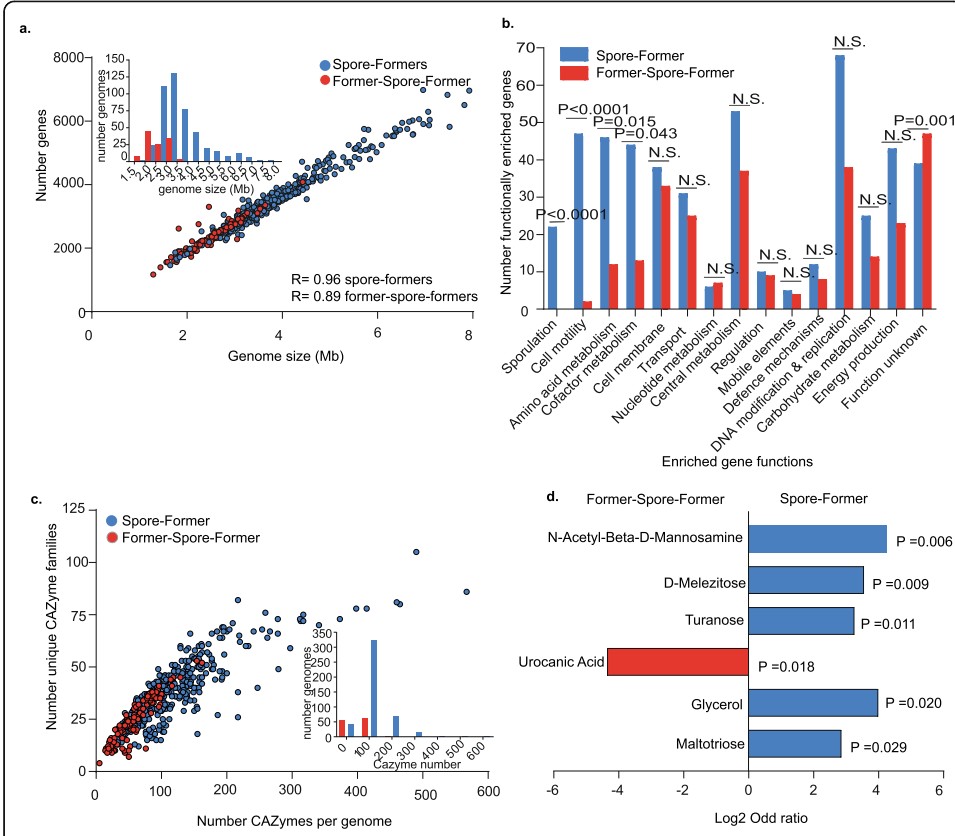

**Fig. 2** Genome reduction and metabolic specialization during host-adaptation by gut Former-Spore-Formers. **a** A marker of host adaptation, genomes of gut FSF are smaller than SF genomes (P< 0.0001, Mann-Whitney test), with a strong correlation between genome size and gene number, Spearman rho, R= 0.96 for SF and R= 0.89 for FSF. Inset shows distribution by genome size. **b** Functional enrichment analysis revealed 489 enriched gut SF genes and 272 enriched FSF genes. Enriched genes were grouped by functional classes. The graph presents the comparison of enriched gene numbers in their functional class and ordered by decreasing statistical significance. Motility, amino acid and cofactor metabolism and sporulation functional classes are statistically more enriched in gut Spore-Formers (SF) compared to Former-Spore-Formers (FSF). No functional classes are more enriched in FSF. Fisher's exact test, N.S. = not significant. **c** FSF encode a smaller number of total CAZymes and a smaller number of CAZyme families per genome compared to gut SF (P<0.0001 for both total number and family number, Welch's t-test). Inset shows the distribution by CAZyme number. **d** *Erysipelotrichaceae* FSF have a more restricted carbohydrate utilization profile compared to *Erysipelotrichaceae* SF. The ability of FSF (n=4) and SF (n=4) to use 95 different carbon sources was tested. N-Acetyl-Beta-D-Mannosamine (P=0.006) (a precursor of sialic acid), D-Melezitose (P=0.009), turanose (P=0.011), glycerol (P=0.020) and maltotriose (P=0.029) are metabolized to a statistically significant greater degree by SF whereas urocanic acid (a derivative of histidine) was metabolized to a statistically significant greater degree by FSF (P=0.018), based on Fisher's exact test

functions in SF may promote stability within the intestinal microbiota by providing essential metabolites [48, 49].

Within FSF, no cobalamin biosynthesis genes are enriched, but two (*BtuB* and *BtuE*) are associated with cobalamin transport. Also, within the cofactor metabolism class, FSF are enriched in 4 genes associated with vitamin K2 (menaquinone) biosynthesis. Interestingly, microbial production of cobalamin is unlikely to benefit human hosts due to an inability to absorb it in the large intestine, unlike menaquinone where absorption is possible [46]. Amino acid and cofactor transport may therefore provide an adaptive efficiency for FSF bacteria avoiding the cost of internal biosynthesis. Thus, for certain

key metabolites, host adaptation by gut bacteria may be characterized as a lifestyle shift from "producer" to "scavenger," potentially promoting colonization of distinct metabolic niches.

### Restricted carbohydrate metabolism in Former-Spore-Formers

Within SF, 25 functionally enriched genes are annotated with carbohydrate metabolism functions compared to 14 in FSF (Fig. 2b). As carbohydrates are the main energy source for gut bacteria, we next annotated the carbohydrate-active enzymes (CAZymes) within Firmicutes gut bacteria. On average, SF genomes have a larger total number of CAZymes and a larger number of different CAZyme families compared to FSF genomes (total number CAZymes: 112 on average per SF genome compared to 57.51 per FSF genome, number CAZymes families: 37 on average per SF genome compared to 24.16 per FSF genome) (total number and family number of CAZyme P<0.0001, Welch's t test) (Fig. 2c). Thus, SF encode a broader repertoire of CAZymes, suggesting a greater saccharolytic ability, which may have been lost in FSF.

The *Erysipelotrichaceae* are a phylogenetically distinct bacterial family within the *Erysipelotrichales* order that remain poorly characterized despite being both health- and disease-associated in humans [50]. Importantly, in our dataset, the *Erysipelotrichaceae* contain multiple gut-associated SF and FSF species Additional file 1: Fig. S5a). We therefore chose to use this family as a model to explore metabolic features of host-adaptation in closely related SF and FSF bacteria residing in the same environment. Reflecting the broader pattern in the Firmicutes (Fig. 2c), *Erysipelotrichaceae* gut SF encode a larger total number and a larger number of CAZyme families compared to *Erysipelotrichaceae* gut FSF (Additional file 1: Fig. S5b) (total number CAZymes: 115 on average per SF genome compared to 57 per FSF genome, number CAZymes families: 34.85 on average per SF genome compared to 23 per FSF genome) (total number and family number of CAZyme P<0.0001 and P= 0.0001, respectively, Welch's t test).

We next used the *Erysipelotrichaceae* to phenotypically validate our genomic analysis results showing a broader carbohydrate metabolic profile in SF. We inoculated phylogenetically diverse bacteria from *Erysipelotrichaceae* SF (n=4) and FSF (n=4) [25, 51] (Additional file 1: Fig. S5a, Additional file 2: Table S3) in Biolog AN MicroPlates containing 95 different diverse carbon sources such as carbohydrates, amino acids, carboxylic acids, and nucleosides. While the AN MicroPlates do not contain the full range of complex carbohydrates targeted by CAZymes, they provide a detailed insight into the metabolic capabilities of isolates tested. Growth was detected with 78 different carbon sources (59 carbon sources for FSF and 69 carbon sources for SF) (Additional file 2: Table S4). When clustered into broad carbon source groups, FSF were more limited in their capacity to utilize both carbohydrates (P<0.0001, Fisher's exact test) and amino acids (P=0.003, Fisher's exact test), consistent with our genomic analysis (Fig. 2b, Fig. 2c, Additional file 1: Fig. S5b).

At the individual carbon source level, FSF also had a reduced metabolic capability compared to SF bacteria (Fig. 2d). Urocanic acid, a derivative of histidine, whose metabolism is linked to short-chain fatty acid production, was the only carbon source

metabolized to a statistically significant greater degree by FSF (P=0.018, Fisher's exact test), suggesting metabolism of specific amino acids present in the intestinal environment may provide colonization-associated advantages in the FSF *Erysipelotrichaceae.* Enriched metabolism of carbon sources by SF include glycerol (P=0.020, Fisher's exact test), which requires cobalamin as a cofactor for its metabolism [52], and N-acetyl-beta-D-mannosamine (β-ManNAc) (P=0.006, Fisher's exact test), a derivative of sialic acid. Thus, we provide evidence of a specialized metabolic capability linked to sporulation loss during host adaptation in Firmicutes gut bacteria.

### Former-Spore-Formers display increased colonization abundance in human populations

Taken together, our genotypic and phenotypic results indicate that the broader metabolic and functional capabilities of SF reflect a more generalist lifestyle, compared to the reduced capabilities of FSF which we propose are adapted to a more stable and specialized lifestyle. We next hypothesized that an inability to make spores in FSF bacteria would limit their environmental survivability and, as a result, reduce their transmission range leading to a lower prevalence in human populations compared to SF bacteria.

To investigate this, we calculated the prevalence of SF and FSF bacteria in 9966 fecal metagenomes representing human adult populations (healthy and disease states) from 6 continents (Additional file 2: Table S5). This was accomplished by reference genome-based mapping to our annotated genomes for SF and FSF bacteria [51]. Importantly, we found that FSF were significantly less prevalent (P=0.0015, two-tailed Wilcoxon rank-sum test) compared to SF (Fig. 3). We obtained the same result when comparing SF and FSF prevalence at the country level, hence the greater prevalence of SF is independent of population-specific factors (P<0.05, two-tailed Wilcoxon rank-sum test) (Additional file 1: Fig. S6). Permutation analysis revealed the higher prevalence of SF was largely driven by a subset of SF with the *Lachnospirales* order (containing *Lachnospiraceae* family) having the most prevalent genomes (65% of *Lachnospirales* genomes are above median prevalence for all SF, 530/1000 permutations using an equal number of SF and FSF genomes resulted in a greater prevalence of SF, P<0.05). These results provide evidence that loss of sporulation limits the transmission range of FSF in human populations.

Our genotypic and phenotypic results indicate that FSF may be more specialized in metabolism and therefore may have a growth advantage in the human gut. We next examined if host adaptation in FSF correlates with an ability to colonize humans to higher levels than SF bacteria. Using reference genome-based mapping, we detected FSF at a significantly higher relative abundance than SF in the human intestinal microbiome (P=0.0034, two-tailed Wilcoxon rank-sum test) (Fig. 3). This was consistent after repeating the analysis with an equal number of SF and FSF genomes (P< 0.05, 1000 permutations). A higher abundance of FSF would promote onward transmission to hosts in close proximity and over short periods, by increasing the levels of excreted bacteria and promoting bacteria maintenance in the local human population. Hence, an absence of sporulation in FSF is correlated with higher abundance levels in the intestinal microbiota, potentially reflecting distinct transmission and colonization strategies in the intestinal microbiota between bacteria that are capable and incapable of sporulation.

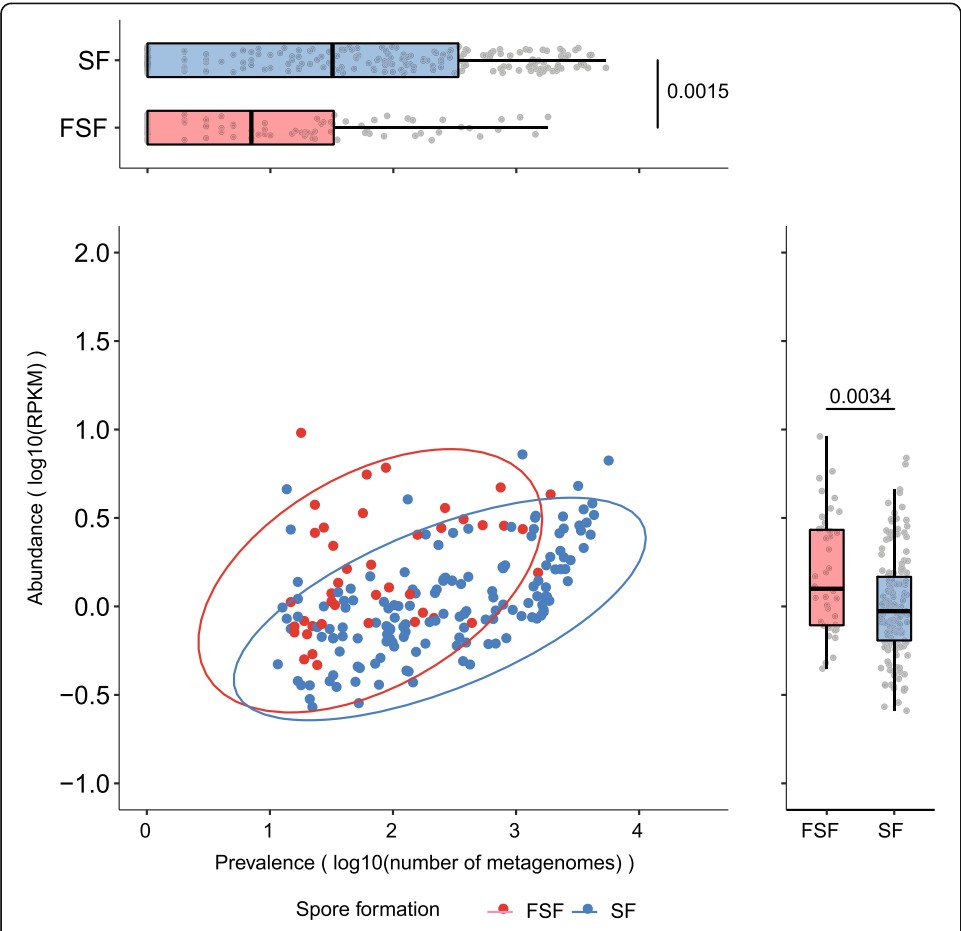

**Fig. 3** Host adaptation is associated with reduced prevalence and higher colonization abundance levels in Former-Spore-Formers. **a** Former-Spore-Formers are less prevalent (P=0.0015, two-tailed Wilcoxon rank-sum test) compared to Spore-Formers within 9966 fecal metagenomes reflecting a reduced transmission range. **b** While less prevalent, Former-Spore-Formers are more abundant (P=0.0034, two-tailed Wilcoxon rank-sum test) compared to Spore-Formers within the same 9966 fecal metagenome**s** reflecting greater host adaptation and an ability to colonize to higher levels. Each dot represents an individual species. Box lengths represent the IQR of the data, and the whiskers the lowest and highest values within 1.5 times the IQR from the first and third quartiles, respectively

## Discussion

Here, we demonstrate that gut Firmicutes commonly lose their ability to make spores during host adaptation. FSF bacteria are less resilient compared to spore-formers, which limits environmental survival, resulting in an altered transmission cycle [7, 25]. Within closely interacting, social groups of baboons, non-spore-forming, anaerobic bacteria (including FSF) are shared to a greater degree than spore-forming bacteria [53], suggesting a transmission cycle that relies on close contact between hosts to limit bacterial exposure to adverse environmental conditions. Furthermore, colonization to high abundance levels which is a feature of Former-Spore-Formers will promote transmission by ensuring high shedding levels in fecal matter that would increase the chances of a successful host colonization event [54]. Indeed, a greater incidence of transmission between mother and infants is observed for FSF compared to SF [19]. Thus, the transmission cycle of FSF bacteria is highly evolved, potentially relying on high-level

colonization abundance to facilitate transmission over short distances and time frames (Fig. 4).

By contrast, the environmental persistence of resilient spores removes the need for direct transmission between hosts in close contact. The larger genomes of SF, encoding a broader metabolic capability, also indicate a more generalist lifestyle adept at surviving in different hosts and environments [55, 56]. Previous studies have shown that human acquisition of SF in early life occurs to a greater degree from environmental sources compared to non-spore-forming bacteria which are maternally acquired [10, 19, 57]. Thus, the transmission cycle of SF relies on the production of resilient spores which increases the proportion of individuals that can potentially be colonized and which is reflected in the greater prevalence of SF in human populations. Hence, SF has a larger transmission range compared to FSF bacteria (Fig. 3).

We also believe the larger transmission range of spore-forming bacteria increases the overall diversity of the human microbiota by providing a source of bacteria capable of sustained gut colonization. We find spore-formers contribute more to beta-diversity (Aitchison distance) compared to non-spore-forming bacteria when examining metagenomes from both within the same country and between different countries (Additional file 1: Fig. S7). Dormancy mechanisms, such as sporulation promote microbial

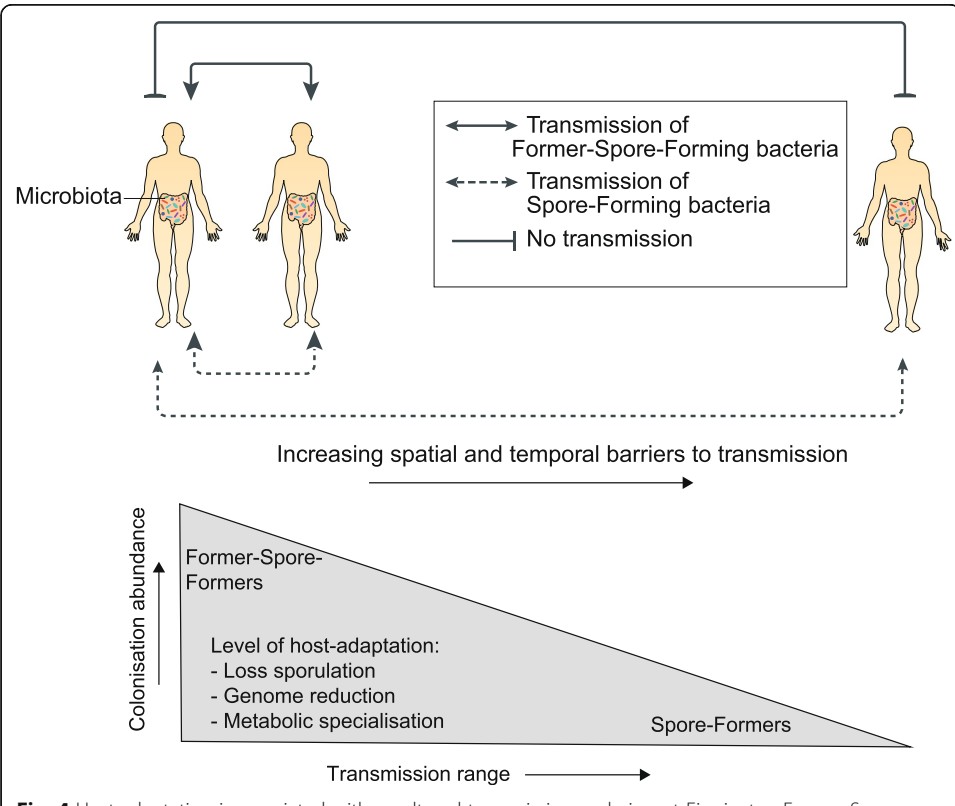

**Fig. 4** Host adaptation is associated with an altered transmission cycle in gut Firmicutes. Former-Spore-Formers (FSF) are more host-adapted compared to Spore-Forming (SF) bacteria as measured by their reduced genome size and genomic redundancy and their more specialist metabolic capabilities. This greater level of host-adaptation corresponds to an ability to colonize to higher abundance levels which promote onward direct transmission to hosts in close proximity. By contrast, SF are less host-adapted and colonize to lower abundance levels. SF transmission cycle relies on the production of resilient spores that promote environmental persistence; thus, they are capable of colonizing a greater proportion of hosts and have a greater prevalence in human populations. Figure adapted from [7]

reservoirs, replenishing species that are lost and occupying newly available niches [56]. Hence, spore-formation may perform an important role in maintaining microbiome stability and functional redundancy as it provides a means for a large number of taxonomically different bacterial species to transmit between hosts.

We propose there is an evolutionary trade-off between high-level colonization abundance mediated by host adaptation and transmission range promoted by sporulation within gut Firmicutes. Sporulation is an energy-expensive biological process requiring synchronization of hundreds of genes, hence it is likely to be lost if no longer needed [32]. Lowly abundant SF bacteria are more at risk of extinction or expulsion from the intestinal microbiota; therefore, sporulation may be maintained as it increases the chances of survival once expelled [58]. Alternatively, loss of sporulation may lead to a different evolutionary trajectory centered on host adaptation, high-level colonization abundance, and a more specialized transmission cycle.

## Conclusion

In this study, we reveal different levels of host adaptation exist within gut Firmicutes, linked to the presence or absence of sporulation. Further studies are required to understand how these adaptive processes shape the functions and ecology of gut bacteria including colonization resistance and microbiota assembly throughout life.

## Methods

### Genomes for analysis

1687 Firmicutes genomes from the NCBI curated RefSeq database (representative genomes), in addition to whole-genome sequences from intestinal isolates from the Human Microbiome Project, a comprehensive study describing the first 1000 intestinal cultured species of the intestinal microbiota and an in-house collection of whole-genome sequences derived from our intestinal bacterial culture collection were used [25, 51, 59–61]. Genomes were annotated using the pipeline described in Page et al. [62]. Redundant genomes were removed and CheckM [63] was then used to filter genomes with less than 90% completeness and greater than 5% contamination leaving 1358 genomes for analysis. Actinobacteria genomes (n=5) from our culture collection were used to root the phylogenetic tree- *Collinsella aerofaciens* (GCA_001406575.1), *Bifidobacterium adolescentis* (GCA_001406735.1), novel *Collinsella* species (GCA_900066465.1), *Collinsella aerofaciens* (GCA_001404695.1), and *B. pseudocatenulatum* (GCA_001405035.1). Bacteroidetes, Proteobacteria, and Actinobacteria genomes for estimation of sporulation ability are described in Additional file 2: Table S1.

### Determination of sporulation ability

We previously identified 66 genes that are predictive for the formation of ethanol-resistant spores using a machine learning approach based on a comparison of genomes derived from 234 bacteria with an ethanol resistant or ethanol sensitive phenotype, cultured from human feces [25]. In this prior study, we applied a strict minimum cut-off of 50% in sporulation signature score (i.e., at least 33 of the 66 sporulation signature score genes present) to classify a genome as capable of sporulation. Here, in this study, using a larger data-set of Firmicutes from different environments (not just the gut), we

counted the number of sporulation signature genes per genome, but instead of using a strict cut-off, we assessed sporulation capability on a taxonomic family by family basis.

Families were first defined using the GTDB database [64] and phylogenetic placement in Fig. 1c. The presence of the 66 sporulation signature genes was then determined using tblastn of the genome sequences against the amino acid sequence of the 66 sporulation signature genes (e value 1e−05 and 30% identity). Genomes in families clustered together with either high sporulation signature scores, low sporulation signature scores, or a bimodal pattern with both high and low scoring signature scores.

*Spo0A* is a DNA-binding protein, whose activation initiates a transcriptional cascade leading to the production of a spore. It is found in all spore-forming bacteria, but its presence per se does not confirm the ability to sporulate [21]. *Spo0A* is also one of the 66 genes in the sporulation signature. The presence or absence of *spo0A* was strongly associated with genomes in high and low scoring clusters, respectively. In total, 98.9% (1343 of 1358) of the genomes are either high scoring for sporulation signature score and contain *spo0A* or are low scoring and lack *spo0A*. Based on this, we classified genomes in high-scoring clusters with *spo0A* as spore-forming and genomes in low-scoring clusters without *spo0A* as incapable of sporulation. If a taxonomic family did not have a bimodal pattern of sporulation signature score, it was classified as high or low scoring based on the presence or absence of *spo0A.* Only 15 genomes (1.1% of total) did not have a high-scoring sporulation signature score and *spo0A* present or a low scoring sporulation score and *spo0A* absent. These 15 genomes have *spo0A*, but as all are part of low scoring sporulation signature clusters, they were classified as Former-Spore-Formers.

This classification system is less stringent than the previously used cut-off of 50% as it classifies some genomes as spore-forming that have a sporulation signature score of less than 50%. However, it accommodates the different sporulation machinery in different taxonomic families. We validated our classification system by generating TEM images (Additional file 1: Fig. S3b), which shows spores for genomes of bacteria predicted to be spore-forming and by literature searches.

### Loss of sporulation genes

To determine loss of sporulation genes in genomes from gut bacteria, the amino acid sequence of the 66 sporulation signature proteins was blasted against whole-genome sequences using tblastn (1e−05, minimum identity 30%). The sporulation signature genes were assigned to specific sporulation stages as previously described [25], and the percentage of genomes containing genes from each sporulation stage was calculated. The number of genes in each stage was stage 0 (3 genes): SF genomes = 1292 and FSF genomes = 288, stage 1 (2 genes): SF genomes = 762 and FSF genomes = 96, stage 2 (7 genes): SF genomes = 1624 and FSF genomes = 134, stage 3 (12 genes): SF genomes = 2165 and FSF genomes = 125, stage 4 (12 genes): SF genomes = 3994 and FSF genomes = 171, stage 5 (9 genes): SF genomes = 2579 and FSF genomes = 130, germination (2 genes): SF genomes = 840 and FSF genomes = 0 and stages unknown (19 genes): SF genomes = 5318 and FSF genomes = 890.

### Phylogenetic analysis

The fetchMG program [61] was used to extract 40 universal genes from the genomes. The resulting amino acid sequences were aligned using mafft (v7.205) [65], and gaps

representing poorly aligned sequences were removed using the Gblocks script (v0.91b) [66] leaving an alignment 6048 amino acids in length. A maximum-likelihood phylogeny was constructed using FastTree [67] (version 2.1.9) using the Jones-Tayler-Thorton (JTT) model of amino-acid evolution and 20 rate categories per site. All bootstrap values using the Shimodaira-Hasegawa test, to at least the family level of the phylogeny are greater than 0.7. This phylogenetic structure is congruent with other large phylogenies such as that implemented in AnnoTree [68] and derived from GTDB [64]. All phylogenies were viewed using iTOL [69]. Habitat origins of isolate genomes were determined using literature searches and available information on NCBI. The *Erysipelotrichaceae* phylogeny was extracted from the main Firmicutes phylogeny.

### Ethanol shock test

Species for the ethanol shock test to validate spore-formation characterization were selected based on phylogenetic diversity (6 different families were tested, *Enterococcaceae*, *Streptococcaceae*, *Lactobacillaceae*, *Erysipelotrichaceae*, *Lachnospiraceae*, and *Peptostreptococcaceae*) and a wide range in sporulation signature scores (36 to 95% for SF and 15 to 29% for FSF). Isolates were streaked from frozen glycerol stocks and then grown overnight in 10ml broth containing YCFA media, a nutrient growth media formulated to culture gut bacteria [37]. Culturing took place in anaerobic conditions in an A95 Whitley Workstation. The next day, the cultures were spun down by centrifugation for 10 min at 4000 rpm to pellets. Ethanol (70% v/v) was added and the pellets were re-suspended and vortexed to ensure complete immersion. Four hours later, the pellets were spun down, ethanol was discarded, and the pellet was washed by immersing in phosphate-buffered saline (PBS), spinning down to obtain a pellet and discarding the PBS. The wash step was repeated and the final pellet was re-suspended in 100mg/ml solution using PBS, serially diluted and plated on YCFA media in anaerobic conditions supplemented with sodium taurocholate to stimulate spore germination. Ethanol resistance was determined by counting colonies (indicating germinated spores) that were present the following day. To account for species that do not sporulate in vitro, if a species was originally cultured from ethanol-treated feces it was considered spore-forming. Species that did not survive ethanol shock treatment were also checked to see if they were originally cultured from non-ethanol treated feces only.

### TEM

Spore images were generated using transmission electron microscopy (TEM) as previously described [70]. Bacterial isolates were streaked from frozen glycerol stocks on YCFA media [37] in anaerobic conditions in an A95 Whitley Workstation, and purity was confirmed by morphological examination and full-length 16S ribosomal RNA gene sequencing. The isolates were then inoculated in YCFA broth for 2 weeks in order to induce stress conditions and stimulate sporulation before TEM images were prepared. Genome accessions of the 6 isolates are No.1 = ERR1022323, No.2= ERR1022375, No. 3= ERR171272, No. 4= ERR1022380, No.5= ERR1022472, and No. 6 = ERR1022333.

### Functional enrichment

To identify protein domains in a genome, RPS-BLAST using conserved protein domains (CDD) database [71] (accessed April 2019) was utilized. Domain and functional enrichment analysis was calculated using one-sided Fisher's exact test with $P$ value adjusted by Hochberg method in R v. 3.2.2. All enriched domains were classified in different functional categories using the COG database (accessed April 2019) and manually curated using the functional scheme originally developed for *Escherichia coli* [72]. In total, 83% of enriched FSF genes (225/272) were assigned to classes of a known function compared to 92% of enriched SF genes (450/489).

### Paralog analysis

To identify paralogs in a genome, protein domains were identified using RPS-BLAST and conserved protein domains (CDD) database [71] (accessed April 2019). Paralogs were called if multiple copies of a protein domain are present in a genome. The percentage of paralogs was calculated using a number of paralogs and the total number of protein domains present in a genome.

### CAZyme analysis

The presence of carbohydrate-active enzymes [73] was determined by querying the dbCAN families in the HMM database using hmmscan against the amino acid sequences of the protein-coding genes in the genomes. Hits were filtered based on an alignment of >80 amino acids using E value of less than 1e−05 or E values of less than 1e−03 covering greater than 30% of the HMM hit. dbCAN families not directly related to carbohydrate utilization were removed prior to analysis, and these were all auxiliary activities, all glycosyltransferases, and carbohydrate esterase 10 (CE10). This left 219 entries in total to query.

### Biolog analysis

The *Erysipelotrichaceae* isolates used for Biolog analysis are described in Additional file 2: Table S3. *Longicatena caecimuris* and *Erysipelatoclostridium ramosum* have been deposited with the NCIMB culture collection under the accessions NCIMB 15236 and NCIMB 15237 respectively as part of this study. All other isolates have been previously isolated by us in the Host-Microbiota Interactions Laboratory and deposited in public culture collections except for *Faecalitalea cylindroides* (DSM3983), *Holdemanella biformis* (DSM3989), and *Eggerthia cateniformis* (DSM20559) which were obtained from DSMZ [74, 75]. The FSF selected have a sporulation signature score ranging from 19–23%, the SF selected have a sporulation signature score ranging from 47 to 56%. Before Biolog experiments, isolates were tested for ethanol resistance or were assessed if originally isolated from ethanol-treated feces. All 4 FSF isolates did not survive ethanol exposure and were not isolated from ethanol-treated feces. For the SF, *Clostridium innocuum* was successfully isolated following ethanol exposure and *Longicatena caecimuris* was not isolated following ethanol exposure but was originally isolated from ethanol-treated feces [25]. *Clostridium spiroforme* and *Erysipeloclostridium ramosum* did not survive ethanol exposure and were not originally isolated from ethanol-treated feces by us. However, there are numerous reports of these two species forming spores

in the literature including imaging [76–78]. Based on this and combined with our genomic predictions, they were characterized as spore-forming.

Isolates were re-streaked on YCFA agar media and grown overnight before using (*Holdemania filiformis* was allowed to grow for 2 days until sufficient growth had occurred). Cotton swabs were used to remove colonies which were then inoculated in AN-IF Inoculating Fluid (Technopath product code 72007) to a turbidity of 65% using a turbidimeter. Then, 100ul was pipetted into each well of Anaerobe AN Microplates (Technopath, product code 1007) which contains 95 different carbon sources. The plates were sealed in PM Gas Bags (Technopath, product number 3032) and run on the Omnilog system for 24 h. For each isolate, 3–5 replicates run on different days from different starting colonies were used. Data was analyzed using the CarboLogR application [79].

### Metagenomic abundance and prevalence

We first determined genome quality of SF and FSF genomes using CheckM [63] ("lineage_wf" function) and then de-replicated at an estimated species-level [80, 81] using dRep v2.2.4 [82]. Briefly, genomes with a Mash [83] distance < 0.1 were first grouped (option "-pa 0.1") and subsequently clustered at an average nucleotide identity of 95% with a minimum alignment fraction of 60% (options "-sa 0.95 -nc 0.60"). The best quality representative genome was selected from each cluster on the basis of the CheckM completeness, contamination, and the assembly N50. Each species representative was taxonomically classified with the Genome Taxonomy Database [64] toolkit v0.2.1 using the "classify_wf" function and default parameters. Sporulation capability was calculated as described above.

To quantify the prevalence and abundance of each species, we aligned the sequencing reads from 28,060 metagenomic datasets to our set of representative species (SF n=258 and FSF n=98) with BWA v0.7.16a-r1181 [84]. The reference database used was first indexed with "bwa index," and metagenomic reads were subsequently aligned with "bwa mem." Prevalence was determined as previously described [85], where species presence was inferred when a genome was covered across at least 60% length, allowing a maximum level of depth variation according to the percentage of the genome covered (taken as the 99th percentile across all data points). Coverage and depth were inferred with samtools v1.5 and the function "depth" [86].

Abundance was quantified by first filtering for uniquely mapped and correctly paired reads ("samtools view -f 2 -q 1") and normalized both by the sample sequencing depth and genome length into Reads Per Kilobase Million (RPKM) using the following formula:

$$RPKM = RS/(GL*TRC/1,000,000)$$

RS represents the number of reads uniquely mapped, GL the reference genome length in kilobases (kb), and TRC the total read count of the metagenomic dataset used for mapping. The level of species prevalence and abundance was compared using a two-tailed Wilcoxon rank-sum test. For estimating differences in abundance, only those species present in more than 10 metagenomic datasets were considered.

## Supplementary Information

---

**Additional file 1: Figure S1**. Environmental distribution of genomes from Firmicutes bacteria. **Figure S2**. Prediction of sporulation capability in Firmicutes. **Figure S3**. Phenotypic validation of sporulation capability predictions. **Figure S4**. Genome reduction and metabolic specialization during host-adaptation by gut Firmicutes. **Figure S5**. Erysipelotrichaceae Former-Spore-Formers have a reduced carbohydrate metabolism profile compared to Erysipelotrichaceae Spore-Formers. **Figure S6**. Former-Spore-Formers are less prevalent than Spore-Formers in gut metagenomes from the same country. **Figure S7**. Spore-formers contribute more to beta-diversity compared to non-spore-forming bacteria in the human intestinal microbiota.

**Additional file 2: Table S1**. Genomes used in this study. **Table S2**. Functionally enriched genes in Spore-Formers and Former-Spore-Formers. **Table S3**. *Erysipelotrichaceae* isolates and culture collection accession numbers used for Biolog experiments. **Table S4**. Carbon sources used by isolates in Biolog experiments. **Table S5**. Metagenomes and metadata used to estimate abundance and prevalence of Spore-Formers and Former-Spore-Formers.

**Additional file 3.** Review history.

---

### Acknowledgements

The authors would like to acknowledge the support of the Wellcome Sanger Institute Pathogen Informatics Team. We thank A. Neville for critical feedback on the manuscript.

### Review history

The review history is available as Additional file 3.

### Peer review information

### Authors' contributions

H.P.B. and T.D.L. conceived the study. H.P.B., N.K, and S.C.F. performed bioinformatics analysis. A.A. performed metagenomics analysis. K.V. analyzed Biolog data. H.P.B, A.T.A., E.V., and N.J.R.D. performed in vitro experiments. C.C. and D.G generated TEMs. H.P.B. and T.D.L. wrote the paper with input from all authors. The authors read and approved the final manuscript.

### Funding

This work was supported by the Wellcome Trust core funding [098051] and the Australian National Health and Medical Research Council [1091097, 1159239, and 1156333 (S.C.F.) and the Victorian Government's Operational Infrastructure Support Program (S.C.F.).
*Funding for open access charge*: Wellcome Sanger Institute.

### Availability of data and materials

Accession numbers of genomes used in this study are listed in Additional file 2: Table S1. Isolates used for Biolog experiments are listed in Additional file 2: Table S3. *Longicatena caecimuris* and *Erysipelatoclostridium ramosum* have been deposited with the NCIMB culture collection under the accessions NCIMB 15236 [87] and NCIMB 15237 [88] respectively as part of this study.

## Declarations

### Ethics approval and consent to participate

Not applicable.

### Consent for publication

Not applicable.

### Competing interests

The authors declare that they have no competing interests.

### Author details

[1]Host-Microbiota Interactions Laboratory, Wellcome Sanger Institute, Hinxton, UK. [2]Wellcome Sanger Institute, Hinxton, UK. [3]European Bioinformatics Institute, Hinxton, UK. [4]Centre for Innate Immunity and Infectious Diseases, Hudson Institute of Medical Research, Clayton, Victoria 3168, Australia. [5]Department of Molecular and Translational Sciences, Monash University, Clayton, Victoria 3800, Australia.

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

## 