## [**Additional file 3.** Review history. · Genome Biology]

Review History

First round of review

Reviewer 1

Are you able to assess all statistics in the manuscript, including the appropriateness of statistical tests used? Yes, and I have assessed the statistics in my report.

Comments to author:

In this paper, Browne et al. investigate the evolution of sporulation in Firmicutes bacteria colonizing gut environments. In particular, they employ genomic and functional experiments to understand the impact that the loss of sporulation in several Firmicutes clades had on genomes, functions and the ecological distribution of bacteria.

Overall, I'm enthusiastic about the paper, and strongly support its publication. The analysis is timely, and leverage the most up-to-date collections of human-associated Firmicutes genomes. I really appreciate the effort of pairing computational analyses with experimental validations, which strengthens the manuscript.

I have a couple of major comments that should be easy to address, and a few minor comments/suggestions listed below.

Major comments

As of now, and unless I'm wrong, all analyses on metabolic differences between FSF and FS bacteria do not seem to be controlling for the effect of phylogeny. I suggest using phylogenetic comparative method(s) to make sure that results on functions being enriched in either FSF or FS bacteria are not resulting from phylogenetic relatedness (which may be at play here, considering that FSF and FS bacteria cluster in major, separate clades).

Section "Former-Spore-Formers display increased colonisation abundance in human populations": the authors only present prevalence data for FSF and FS bacteria at the aggregate level. It could be possible that the trend being observed is not due to sporulation but is confounded by other host-related or population-specific factors that would favor the prevalence of SF bacteria in the microbiome. Can the authors replicate their results when running analyses at lower resolutions, e.g. within host populations?

Minor comment

L233-251: This section is about experimentally validating the computational prediction of a larger spectrum of carbohydrate metabolic capacities among spore-former bacteria, based on the distribution of CAZyme families. For this, the authors chose to screen for the metabolism of simple carbon sources using a Biolog MicroPlate. While a few CAZyme families are involved in the processing of simple sugars, many are involved in the degradation of complex sugars. Could the authors comment on this and acknowledge the limitations of their experimental validation? Also, the discussion on amino acids hasn't been motivated, and does not seem to validate genomic analyses as claimed at the moment (L250-251)?

Additional comments

L93: "different human different body sites"

L94-95: what's the rationale for selecting the 72 non-Firmicutes genomes/bacteria?

L103-107: the results being discussed here should be referenced - in what figure/table these results are being shown?

L124: please be more explicit about what is being tested here with the Mann-Whitney test

L154-155: maybe clarify that TEM has been done on a subset of bacteria? Or has it been done for all but only data for 6 species are shown in FigS3b?

L213: replace '2' with 'two'

Fig2a: there's a ' $P < 0.0001$ ' below this panel that is misplaced.

L461: was a criterion for alignment length used? Coverage length between query and subject sequences should be used to more accurately capture homologous gene sequences (and remove hits due to shared domains).

L545-549: do you mean protein coding genes (rather than protein domains)?

Reviewer 2

Are you able to assess all statistics in the manuscript, including the appropriateness of statistical tests used? No, I do not feel adequately qualified to assess the statistics.

Comments to author:

The manuscript by Browne and colleagues reports an analysis on the prevalence of sporulation genes within the phylum Firmicutes, with a particular focus on the evolution of this trait within gut bacteria. The authors conduct a series of large in silico analyses that lead them to describe an association between the loss of sporulation capacity, genome reduction and adaptation to the host environment, as reflected by a narrower capacity to utilize a range of carbon and amino acid sources as opposed to spore formers. The authors also explored the presence of spore formers within gut Firmicutes on human gut metagenome datasets, leading them to establish that non-spore formers, while less prevalent across populations, reach higher relative abundances within individuals. They argue that these facts suggest that both spore and non-spore formers utilize different strategies for human-to-human transmission, with non spore formers presenting a narrower transmission range.

In my opinion, the work is well presented, interesting, and could be of outstanding interest for a broad readership, and particularly for those engaged in human microbiota research, however most of the data presented comes exclusively from in silico analyses and there is little experimental validation of the conclusions obtained from them. An experimental evolution trial would have been the ideal approach to further support the soundness of the conclusions although I understand this might not be an easy task. I suggest the authors could at least better describe the rationale that led them to select exclusively these eight bacteria for phenotypic validation, and discuss how well their conclusions might extend to other families. Also, is there any data supporting if the conclusions of the study might be also true for other phyla?

In addition the work and conclusions rely on the assumption that sporulation is a trait that has

exclusively evolved through gene loss within the phylum Firmicutes, and while this might be the dominant evolution pattern for this trait due to the large number of genes it requires, there is no mention at all about the possibility that it might have also spread through horizontal gene transfer, at least in some cases. Furthermore, genome reduction is a well established evolution pattern, particularly in bacteria inhabiting the human gut, thus I don't think it represents, by itself, a major novel result. I suggest the authors might include some information on these facts in the document.

Also I miss some information on the implications that the conclusions from this work might have on human microbiota research. For instance, would the loss of sporulation likely contribute to the appearance of larger differences in the healthy gut microbiota composition across distant populations?

Finally, could the authors verify that the colours in Fig S5b are correct? They appear discordant with the description of the figure in the main text and legend, with a higher abundance of CAZy families in FSF.

Reviewer 1:

Reviewer #1: In this paper, Browne et al. investigate the evolution of sporulation in Firmicutes bacteria colonizing gut environments. In particular, they employ genomic and functional experiments to understand the impact that the loss of sporulation in several Firmicutes clades had on genomes, functions and the ecological distribution of bacteria.

Overall, I'm enthusiastic about the paper, and strongly support its publication. The analysis is timely, and leverage the most up-to-date collections of human-associated Firmicutes genomes. I really appreciate the effort of pairing computational analyses with experimental validations, which strengthens the manuscript.

We thank the reviewer for their supportive words. Below, we have responded point-by-point to your comments and suggestions. We thank you for your time in reviewing our manuscript.

I have a couple of major comments that should be easy to address, and a few minor comments/suggestions listed below.

Major comments

1. As of now, and unless I'm wrong, all analyses on metabolic differences between FSF and FS bacteria do not seem to be controlling for the effect of phylogeny. I suggest using phylogenetic comparative method(s) to make sure that results on functions being enriched in either FSF or FS bacteria are not resulting from phylogenetic relatedness (which may be at play here, considering that FSF and FS bacteria cluster in major, separate clades).

Using functional enrichment analysis, we aimed to identify gene functions associated with loss or presence of sporulation that could provide insights into the ecology of these gut bacteria. We believe the greater number of functions enriched in SF reflects a more generalist lifestyle, that combined with their larger genomes, greater genetic redundancy and greater metabolic capability (as demonstrated using Biolog AN MicroPlates) permit spore-formers to colonise a larger proportion of human populations or even to survive in different hosts (as has been demonstrated for spore-forming *Clostridioides difficile* that can transmit between farm animals and humans -PMID: 29237792). We do not assume that presence or loss of sporulation is driving these enriched gene functions, rather we highlight associations which we describe in the text, e.g line 204 “Thus, loss of these functions may be linked to loss of sporulation”.

To explore this point further, we carried out the same functional enrichment analysis using the *Erysipelotrichaceae* as they contain both SF and FSF that are phylogenetically related (in the same family) and that are found in the same gut environment. The number of genomes are too small (SF=40 and FSF=10) to achieve statistical significance after correcting for multiple comparisons which is why we didn't include this analysis in the results, however, examining the functions of genes enriched before correcting for multiple comparisons we do see a similar trend to the main SF vs FSF comparison.

In total, there are 116 genes enriched in *Erysipelotrichaceae* gut SF and none in *Erysipelotrichaceae* FSF ($P < 0.05$, not significant after correcting for multiple comparisons using Hochberg method). All functions have a greater number of genes in SF compared to FSF. In addition, we observe the same functional categories such as “sporulation” (n=27), “Amino acid metabolism” (n=12), and “cofactor

metabolism” (n=5). Within “Amino acid metabolism”, 10 of the 12 genes are associated with biosynthesis.

Hence, within the same family, we see the same pattern of gene enrichment as we reported in the analysis of all SF and all FSF in the gut. This suggests our functional enrichment results are connected to presence or absence of sporulation and are not solely due to the phylogeny of the bacteria analysed.

2. Section "Former-Spore-Formers display increased colonisation abundance in human populations": the authors only present prevalence data for FSF and FS bacteria at the aggregate level. It could be possible that the trend being observed is not due to sporulation but is confounded by other host-related or population-specific factors that would favor the prevalence of SF bacteria in the microbiome. Can the authors replicate their results when running analyses at lower resolutions, e.g. within host populations?

This is an interesting point which we have now investigated. We have now conducted this analysis at the country level (only considering countries with >150 samples). Supporting our initial results, SF are more prevalent than FSF for all 9 countries tested (and achieves statistical significance for 8 out of 9 using two-tailed Wilcoxon rank-sum test). We have included this analysis as Supplemental Figure 6 (see below) and **Line 285** “We obtained the same result when comparing SF and FSF prevalence at the country level, hence the greater prevalence of SF is independent of population-specific factors (P<0.05, two-tailed Wilcoxon rank-sum test) (**Additional File 1: Fig. S6**).”

Figure S6

Figure S6: Former-Spore-Formers are less prevalent than Spore-Formers in gut metagenomes from the same country

Former-Spore-Formers (red) are less prevalent compared to Spore-Formers (blue) in gut metagenomes from the same country ($P < 0.05$, two-tailed Wilcoxon rank-sum test for 8 of 9 countries tested). Only countries with more than 150 samples were included in the analysis. Each dot represents an individual species. Box lengths represent the IQR of the data, and the whiskers the lowest and highest values within 1.5 times the IQR from the first and third quartiles, respectively.

Minor comment:

1. L233-251: This section is about experimentally validating the computational prediction of a larger spectrum of carbohydrate metabolic capacities among spore-former bacteria, based on the distribution of CAZyme families. For this, the authors chose to screen for the metabolism of simple carbon sources using a Biolog MicroPlate. While a few CAZyme families are involved in the

processing of simple sugars, many are involved in the degradation of complex sugars. Could the authors comment on this and acknowledge the limitations of their experimental validation?

We used the Biolog AN MicroPlates as they contain 95 different carbon sources representing a wide range of different nutrients present in the gut such as carbohydrates, amino acids, sugar alcohols, nucleosides and carboxylic acids. Therefore, while these plates do not contain the full diversity of complex carbohydrates present in the human gut, they do provide a high throughput means to screen gut bacteria for usage of a wide range of different nutrients.

We believe the Biolog results showing a reduced metabolic capability in *Erysipelotrichaceae* FSF support the other results that indicate host adaptation in FSF such as reduced genome size, reduced genetic redundancy and lower number of enriched functions (including carbohydrate usage).

We now explain better in the text the capabilities and limitations of the AN MicroPlates, (Line 255): “We inoculated phylogenetically diverse bacteria from *Erysipelotrichaceae* SF (n=4) and FSF (n=4) (25, 49) (Additional File 1: Fig. S5a, Additional File 4: Table S3) in Biolog AN MicroPlates containing 95 different diverse carbon sources such as carbohydrates, amino acids, carboxylic acids and nucleosides. While the AN MicroPlates do not contain the full range of complex carbohydrates targeted by CAZymes, they provide a detailed insight into the metabolic capabilities of isolates tested”.

2. Also, the discussion on amino acids hasn't been motivated, and does not seem to validate genomic analyses as claimed at the moment (L250-251)?

Amino acid metabolism is enriched in SF genomes compared to FSF genomes (Figure 2b). We have now referenced this figure in the text (Line 258). “When clustered into broad carbon source groups, FSF were more limited in their capacity to utilise both carbohydrates ($P < 0.0001$, Fisher’s exact test) and amino acids ($P = 0.003$, Fisher’s exact test), consistent with our genomic analysis (Fig 2b, Fig. 2c, Additional File 1: Fig. S5b).”

Additional comments:

1. L93: "different human different body sites"

We have now corrected this (Line 92).

2. L94-95: what's the rationale for selecting the 72 non-Firmicutes genomes/bacteria?

The 72 genomes from Bacteroidetes, Actinobacteria and Proteobacteria were selected as they represent species that are found in the human intestinal microbiota. We have now added the species name for these 72 genomes in Supplemental Table 1 to provide more context for the reader.

3. L103-107: the results being discussed here should be referenced - in what figure/table these results are being shown?

These results refer to Figure 1a and Supplemental Table 1. We have now referenced these in the text (Line 107).

4. L124: please be more explicit about what is being tested here with the Mann-Whitney test

Here, we tested the difference in sporulation signature score between genomes of Spore-Formers, Former-Spore-Formers and non-Firmicutes non-spore-formers (Bacteroidetes, Actinobacteria and Proteobacteria). We have now clarified this in the text (Line 124). “Furthermore, other bacteria from

the Bacteroidetes, Actinobacteria and Proteobacteria phyla which do not make spores are also classified as FSF ($P < 0.0001$, Mann-Whitney, sporulation signature score comparison between genomes of SF, FSF and non-Firmicutes)”

5. *L154-155: maybe clarify that TEM has been done on a subset of bacteria? Or has it been done for all but only data for 6 species are shown in FigS3b?*

We have now clarified this (**Line 156**). It now reads “Transmission Electron Microscopy (TEM) imaging of 21 of the 41 species confirmed the presence of spores in spore-forming bacteria only. TEM images of spores from six species representing four different bacterial families are shown in (**Additional File 1: Fig. S3b**).”

6. *L213: replace '2' with 'two'*

We have now edited this (**Line 218**).

7. *Fig2a: there's a 'P<0.0001' below this panel that is misplaced.*

We have now removed this.

8. *L461: was a criterion for alignment length used? Coverage length between query and subject sequences should be used to more accurately capture homologous gene sequences (and remove hits due to shared domains).*

We did not use a threshold for alignment length, however we have found the BLAST thresholds used (e-value $1e-05$ and 30% identity) to be robust as they corroborate our phenotypic results for isolating spore-formers based on culturing after ethanol exposure, i.e. bacteria characterised as spore-formers based on the number of sporulation signature genes in their genomes were only isolated from ethanol treated samples and bacteria characterised as incapable of sporulation were only isolated from non-ethanol treated samples. Furthermore, spores were only visible in TEM images of bacteria characterised as spore-formers.

9. *L545-549: do you mean protein coding genes (rather than protein domains)?*

We have now corrected these methods. The conserved protein domains (CDD) database not COG was used to do the functional annotation. This also applies to the paralogs analysis. These methods now read as follows (**Line 577**) :

Functional enrichment:

To identify protein domains in a genome, RPS-BLAST using conserved protein domains (CDD) database (71) (accessed April 2019) was utilised. Domain and functional enrichment analysis was calculated using one-sided Fisher's exact test with P value adjusted by Hochberg method in R v. 3.2.2. All enriched domains were classified in different functional categories using the COG database (accessed April 2019) and manually curated using the functional scheme originally developed for Escherichia coli (72). In total, 83% of enriched FSF genes (225/272) were assigned to classes of a known function compared to 92% of enriched SF genes (450/489).

Paralog analysis:

To identify paralogs in a genome, protein domains were identified using RPS-BLAST and conserved protein domains (CDD) database (71) (accessed April 2019). Paralogs were called if multiple copies of a protein domain are present in a genome. Percentage of paralogs was calculated using number of paralogs and total number of protein domains present in a genome.

Reviewer #2:

The manuscript by Browne and colleagues reports an analysis on the prevalence of sporulation genes within the phylum Firmicutes, with a particular focus on the evolution of this trait within gut bacteria. The authors conduct a series of large in silico analyses that lead them to describe an association between the loss of sporulation capacity, genome reduction and adaptation to the host environment, as reflected by a narrower capacity to utilize a range of carbon and amino acid sources as opposed to spore formers. The authors also explored the presence of spore formers within gut Firmicutes on human gut metagenome datasets, leading them to establish that non-spore formers, while less prevalent across populations, reach higher relative abundances within individuals. They argue that these facts suggest that both spore and non-spore formers utilize different strategies for human-to-human transmission, with non spore formers presenting a narrower transmission range.

In my opinion, the work is well presented, interesting, and could be of outstanding interest for a broad readership, and particularly for those engaged in human microbiota research, however most of the data presented comes exclusively from in silico analyses and there is little experimental validation of the conclusions obtained from them. An experimental evolution trial would have been the ideal approach to further support the soundness of the conclusions although I understand this might not be an easy task. I suggest the authors could at least better describe the rationale that led them to select exclusively these eight bacteria for phenotypic validation, and discuss how well their conclusions might extend to other families. Also, is there any data supporting if the conclusions of the study might be also true for other phyla?

We thank the reviewer for their supportive words and for taking the time to review our manuscript.

We selected these 8 bacterial species (4 spore-formers and 4 former-spore-formers) as they all reside within the same bacterial family (Erysipelotrichaceae) and are all found in the gut. This removes any confounders associated with using bacteria from different host environments or diverse taxa and increases our confidence that metabolic differences are due to presence or absence of sporulation.

We believe the Biolog results are applicable to other families especially when we consider the other results that support our conclusion of greater host-adaptation in Former-Spore-Formers such as reduced genome size, reduced genetic redundancy and smaller number of CAZymes (which we observed in the *Lachnospiraceae* and *Peptostreptococcaceae*, as well as *Erysipelotrichaceae*). However, outside of the *Erysipelotrichaceae*, no other family had multiple spore-forming and former-spore-forming gut species for us to test.

As sporulation is only present in the Firmicutes, we cannot say how transmission strategies impact host adaptation in other phyla. We feel this is beyond the scope of our study.

To clarify our selection of *Erysipelotrichaceae*, we have added to the text (Line 241): “We therefore chose to use this family as a model to explore metabolic features of host-adaptation in closely-related SF and FSF bacteria residing in the same environment.”

In addition the work and conclusions rely on the assumption that sporulation is a trait that has exclusively evolved through gene loss within the phylum Firmicutes, and while this might be the dominant evolution pattern for this trait due to the large number of genes it requires, there is no mention at all about the possibility that it might have also spread through horizontal gene transfer, at least in some cases. Furthermore, genome reduction is a well established evolution pattern, particularly in bacteria inhabiting the human gut, thus I don't think it represents, by itself, a major novel result. I suggest the authors might include some information on these facts in the document.

Our results support loss of sporulation and not gain as the evolutionary trend for this phenotype. Considering the hundreds of genes required to make a spore and the major morphological changes involved, it seems highly unlikely that sporulation can be acquired in its entirety through horizontal gene transfer. Sporulation is believed to have evolved once in an early ancestor of the Firmicutes which is reported in several studies (Galperin Microbiology Spectrum 2013, Abecasis *et al* Journal of Bacteriology 2013, Ramos-Silva *et al*, Molecular Biology and Evolution 2019) and which we refer to in **Line 113**: “As bacterial sporulation is believed to have evolved once, early in Firmicutes evolution (21, 35, 36)”...

In the Ramos-Silva study they show a major acquisition of sporulation genes including the essential master regulator *spo0A* at the origin of the Firmicutes. A second major gain is reported in an early *Bacilli* but this is believed to contribute to the diversity in sporulation machinery between *Clostridia* and *Bacilli* classes and not as a means of sporulation acquisition. Therefore, while sporulation-associated genes can be acquired through horizontal gene transfer, there is no evidence to suggest that the sporulation phenotype itself can be acquired.

Regarding genome reduction, we agree with the reviewer that it is commonly observed, especially as a feature of host adaptation. In our study, we link genome reduction as a feature of host adaptation with loss of sporulation but we did not intend to put forward that the genome reduction we observe in gut bacteria is a novel result in itself. To clarify this, we have amended the text (**Line 181**) “Genome reduction is a feature of host-adaptation that has been observed in different environments, **including the human gut** and is characterised by a loss of genes not required to survive in an ecosystem (40-45).

Also I miss some information on the implications that the conclusions from this work might have on human microbiota research. For instance, would the loss of sporulation likely contribute to the appearance of larger differences in the healthy gut microbiota composition across distant populations?

This is an interesting point which we have now expanded on in the discussion. We believe the greater prevalence of spore-formers could explain some of the variability in microbiome composition observed in different populations. Spore-formers are taxonomically diverse with many different species, especially compared to non-spore-forming bacteria such as *Bacteroides*. We compared the beta diversity (Aitchison distance) between metagenomes from the same or different countries when only considering spore-forming or non-spore-forming species. Spore-formers contribute more to the beta-diversity for both comparisons (new supplementary Fig S7, see below). This indicates the ability of spore-formers to readily transmit may contribute to overall diversity in the intestinal microbiota.

We have added text as follows to the discussion (**Line 330**) “We also believe the larger transmission range of spore-forming bacteria increases the overall diversity of the human microbiota by providing a source of bacteria capable of sustained gut colonisation. We find spore-formers contribute more to beta-diversity (Aitchison distance) compared to non-spore-forming bacteria when examining metagenomes from both within the same

country and between different countries (Fig S7). Dormancy mechanisms, such as sporulation promote microbial reservoirs, replenishing species that are lost and occupying newly available niches (56). Hence, spore-formation may perform an important role in maintaining microbiome stability and functional redundancy as it provides a means for a large number of taxonomically different bacterial species to transmit between hosts”.

Figure S7: Spore-formers contribute more to beta-diversity compared to non-spore-forming bacteria in the human intestinal microbiota.

Beta-diversity (Aitchison distance) of metagenomes was calculated for spore-forming and non-spore-forming bacterial species both within the same country and between different countries. Spore-forming species contribute more to beta-diversity than non-spore-forming bacteria (two-tailed Wilcoxon rank-sum test).

Finally, could the authors verify that the colours in Fig S5b are correct? They appear discordant with the description of the figure in the main text and legend, with a higher abundance of CAZy families in FSF.

We have now corrected the colours in this figure- blue should be Spore-Formers and red should be Former-Spore-Formers.

Second round of review

Reviewer 1

The authors have satisfactorily addressed my points. I have no further comments to make on the manuscript and support its publication.